# Noncommutative Correction to the Entropy of Charged BTZ Black Hole

**Tajron Jurić** * and **Filip Požar**

Rudjer Bošković Institute, Bijenička c.54, HR-10002 Zagreb, Croatia
* Correspondence: tjuric@irb.hr

**Abstract:** Noncommutative geometry is an established potential candidate for including quantum phenomena in gravitation. We outlined the formalism of Hopf algebras and its connection to the algebra of infinitesimal diffeomorphisms. Using a Drinfeld twist, we deformed spacetime symmetries, algebra of vector fields and differential forms, leading to a formulation of noncommutative Einstein equations. We studied a concrete example of charged BTZ spacetime and deformations steaming from the so-called angular twist. The entropy of the noncommutative charged BTZ black hole was obtained using the brick-wall method. We used a charged scalar field as a probe and obtained its spectrum and density of states via WKB approximation. We provide the method used to calculate corrections to the Bekenstein–Hawking entropy in higher orders in WKB, but we present the final result in the lowest WKB order. The result is that, even in the lowest order in WKB, the entropy, in general, contains higher powers in $\hbar$, and it has logarithmic corrections and polynomials of logarithms of the black hole area.

**Keywords:** black hole entropy; noncommutative gravity; brick wall

## 1. Introduction

The quest to understand the quantum aspects of gravity has a long history. Namely, since Einstein introduced the theory of general relativity (GR) [1] and the first black hole (BH) solution by Schwarzschild [2], there has been much effort and many research directions that have the goal to formulate some sort of quantum gravity theory. In the center of most of them is the study of quantum aspects of BHs, with entropy being the most interesting physical quantity.

The entropy of BHs was first understood by Bekenstein [3] and Hawking [4] in the framework of semi-classical gravity. Later, it was argued that the origin of BH entropy is rooted in the statistical mechanics of in-falling particles [5] after a proper regularization of the UV divergence [6,7]. There are many approaches (string theory [8–10], quantum geometry [11], conformal field theory [12–16], AdS/CFT duality [17], effective field theory [18–22] and noncommutative geometry [23–25], to name a few) toward quantum gravity in which the entropy of BHs coincides with the Bekenstein–Hawking area law in the leading order.

In this paper, motivated by the possibility of the co-existence of GR and Heisenberg's uncertainty principle [26,27], we used the noncommutative (NC) geometry framework as the underlying theory that captures quantum aspects of gravity [28–32]. It is natural to expect that, at very small scales, i.e., the Planck scale, the structure of smooth manifolds leading to the classical spacetime structure of GR will be modified in any theory of quantum gravity and therefore has to be replaced with some other structure. This new structure is the NC space, represented by some NC algebra and a corresponding NC differential geometry that will enable a formulation of an NC gravity theory [33,34]. In [26,27], it was explicitly shown that the principles of GR together with the Heisenberg's uncertainty principle lead to an NC structure of spacetime.

There have been various attempts to construct a well-defined NC gravity theory [35–46], where it has been shown [47,48] that the NC version of the BTZ black hole [49] is related

to $\kappa$-deformed algebra [50–55]. In addition, a similar example was found for the NC generalization of a Kerr black hole [56] and in some version of NC cosmology [38], leading to the fact that $\kappa$-deformed algebras have an intriguing connection, a sort of universality, when investigating quantum aspects of BHs. Therefore, in this paper, we dealt with a special type of $\kappa$-deformation that comes from a Drinfeld twist [57–61].

The charged BTZ (QBTZ) black hole is a solution of Einstein–Maxwell equations in 2+1 dimensions [62–65]. It is a 2+1-dimensional generalization/simplification of Reissner–Nordström–AdS black hole [66]. Since gravity in 2+1 dimensions has no propagating degrees of freedom and, in principle, can be quantized [67,68], it provides a perfect laboratory for analyzing quantum aspects of gravity and, in particular, the entropy of BHs. This, together with the fact that an angular twist renders NC corrections only to the coupling between the electromagnetic potential and charged scalar field while keeping the background classical [58–61], is our main motivation for studying the entropy of an NC QBTZ.

The paper is organized as follows. In Section 2, we first outline the Hopf algebra approach to the symmetries of manifolds and then present its deformation via the Drinfeld twist leading to a formulation of NC gravity theory. Using the angular twist, we show that QBTZ is a solution of NC Einstein equations and we derive NC corrections to the equation of motion for a charged scalar field. In Section 3, the brick wall method for calculating the entropy of BHs is outlined in general and then used to calculate the quantum and NC corrections to the entropy of QBTZ BH. In Section 4, we study certain limiting cases and finally, in Section 5, we conclude the paper with some final remarks.

Throughout the paper, we use units where $k_B = c = G = 1$, but we explicitly write $\hbar$ since we use WKB and expansions in $\hbar$.

## 2. Noncommutative QBTZ

In this section, we will first motivate the use of Hopf algebra when dealing with infinitesimal diffeomorphisms, and then introduce the Drinfeld twist and show how it generates NC differential geometry, ultimately leading to a formulation of NC gravity. Using this formalism, we will analyze the deformation of the QBTZ-metric via the angular twist and derive the NC generalization of the equation of motion for the charged scalar field.

### 2.1. Infinitesimal Diffeomorphisms and Hopf Algebra

The symmetries of a manifold $\mathcal{M}$ are encoded into the algebra of infinitesimal diffeomorphisms. The infinitesimal diffeomorphism generated by some smooth vector field $v$ acts on the algebra of smooth functions on $\mathcal{M}$, i.e., $\mathcal{C}^\infty(\mathcal{M})$, as Lie derivatives

$$\delta_v f = \mathcal{L}_v(f) = v(f) = v^\mu \partial_\mu f, \quad \forall f \in \mathcal{C}^\infty(\mathcal{M}) \tag{1}$$

where, in the last equality, we used a local basis $\{\partial_\mu\}$ of smooth vector fields $\Xi(\mathcal{M})$. The algebra of infinitesimal diffeomorphisms is described by the Lie algebra of vector fields, where the Lie bracket $[\cdot, \cdot] : \Xi(\mathcal{M}) \times \Xi(\mathcal{M}) \longrightarrow \Xi(\mathcal{M})$ is given by

$$[v, w] = u, \quad v, w, u \in \Xi(\mathcal{M}) \tag{2}$$

where, in the local basis $\{\partial_\mu\}$, we have

$$v = v^\mu \partial_\mu, \quad w = w^\mu \partial_\mu, \quad u = \left( v^\alpha \frac{\partial w^\mu}{\partial x^\alpha} - w^\alpha \frac{\partial v^\mu}{\partial x^\alpha} \right) \partial_\mu \tag{3}$$

and $v^\mu, w^\mu \in \mathcal{C}^\infty(\mathcal{M})$. The Lie algebra of vector fields $(\Xi(\mathcal{M}), [\cdot, \cdot])$ together with its dual, i.e., one-forms $\Omega^1(\mathcal{M})$, plays a central role in differential geometry, and its action on any tensor field is represented via the Lie derivative due to the compatibility identity

$$\mathcal{L}_v \mathcal{L}_w - \mathcal{L}_w \mathcal{L}_v = \mathcal{L}_{[v,w]}, \quad \forall v, w \in \Xi(\mathcal{M}). \tag{4}$$

Note that, for any vector field $v \in \Xi(\mathcal{M})$, there exists $v_{inv} = -v$, which generates the inverse infinitesimal diffeomorphism. In addition, the action of Lie algebra $(\Xi(\mathcal{M}), [\cdot, \cdot])$ on any tensor product between tensors $T$ and $T'$, i.e., the product of representations $T \otimes T'$, is defined by the Leibniz rule

$$\mathcal{L}_v(T \otimes T') = \mathcal{L}_v(T) \otimes T' + T \otimes \mathcal{L}_v(T'). \tag{5}$$

If we now introduce a canonical map $\epsilon : \Xi(\mathcal{M}) \longrightarrow \mathbb{C}$, such that it associates to each vector field the number zero, $v \mapsto 0$, the structure of a Hopf algebra emerges.

A Hopf algebra $H$ is an algebra (also a co-algebra) together with three maps: coproduct $\Delta$, counit $\epsilon$ and antipod $S$, that satisfy

$$(\Delta \otimes 1)\Delta = (1 \otimes \Delta)\Delta, \tag{6}$$

$$(\epsilon \otimes 1)\Delta = 1 = (1 \otimes \epsilon)\Delta, \tag{7}$$

$$m[(S \otimes 1)\Delta] = \epsilon = m[(1 \otimes \epsilon)\Delta], \tag{8}$$

where $m(a \otimes b) = ab, \forall a, b \in H$ is the algebra multiplication. In our example, the Lie algebra of vector fields $(\Xi(\mathcal{M}), [\cdot, \cdot])$, the Hopf algebra is given by $H = (\mathcal{U}\Xi, m, \Delta, \epsilon, S)$, where $\mathcal{U}\Xi$ is the universal enveloping algebra of the Lie algebra $(\Xi(\mathcal{M}), [\cdot, \cdot])$, $m$ is the product in $\mathcal{U}\Xi$ (namely the composition of vector fields) and $\Delta : \mathcal{U}\Xi \longrightarrow \mathcal{U}\Xi \otimes \mathcal{U}\Xi$ is a $\mathbb{C}$-linear homomorphism that encodes the Leibniz rule (5) and is given by

$$\Delta(v) = v \otimes 1 + 1 \otimes v, \quad \Delta(1) = 1 \otimes 1, \quad \forall v \in \Xi(\mathcal{M}), \tag{9}$$

$S : \mathcal{U}\Xi \longrightarrow \mathcal{U}\Xi$ is a $\mathbb{C}$-linear antihomomorphism that encodes the algebra inverse

$$S(v) = -v, \quad S(1) = 1, \quad \forall v \in \Xi(\mathcal{M}), \tag{10}$$

and $\epsilon : \mathcal{U}\Xi \longrightarrow \mathbb{C}$ is a $\mathbb{C}$-linear map that encodes the normalization

$$\epsilon(v) = 0, \quad \epsilon(1) = 1 \quad v \in \Xi(\mathcal{M}). \tag{11}$$

In conclusion, one says that the symmetries of a manifold $\mathcal{M}$ are fully encoded into its corresponding Hopf algebra $H = (\mathcal{U}\Xi, m, \Delta, \epsilon, S)$.

### 2.2. Deformed Symmetries, Twist and Noncommutativity

The Hopf algebra framework is suitable for investigating quantum or, more specifically, deformed symmetries [69]. One of the most comprehensive ways of realizing this is by using the concept of a Drinfeld twist [70–72]. A Drinfeld twist (here, we need to extend the notion of universal enveloping algebra to formal power series in parameter $\lambda$. We replace the field $\mathbb{C}$ with the ring $\mathbb{C}[[\lambda]]$, so we promote $\mathcal{U}\Xi \longrightarrow \mathcal{U}\Xi[[\lambda]]$, but, for brevity, we drop this notation (for more details see [34,69])) $\mathcal{F} \in \mathcal{U}\Xi \otimes \mathcal{U}\Xi$ is an invertible element that satisfies (we also demand $\mathcal{F} = 1 \otimes 1 + \mathcal{O}(\lambda)$)

$$\mathcal{F} \otimes 1(\Delta \otimes 1)\mathcal{F} = 1 \otimes \mathcal{F}(1 \otimes \Delta)\mathcal{F} \quad \text{cocycle condition} \tag{12}$$

$$(\epsilon \otimes 1)\mathcal{F} = 1 = (1 \otimes \epsilon)\mathcal{F} \quad \text{normalization condition} \tag{13}$$

Provided that a Drinfeld twist exists, there is a well-known theorem that enables us to construct a new Hopf algebra $H_{\mathcal{F}} = (\mathcal{U}\Xi, m, \Delta_{\mathcal{F}}, \epsilon, S_{\mathcal{F}})$, where

$$\Delta_{\mathcal{F}} = \mathcal{F}\Delta\mathcal{F}^{-1} \tag{14}$$

$$S_{\mathcal{F}} = \alpha S \alpha^{-1}, \quad \alpha = m[\mathcal{F}(1 \otimes S)] \tag{15}$$

This new Hopf algebra $H_{\mathcal{F}}$ describes the deformed infinitesimal diffeomorphisms, i.e., the symmetry of the deformed or noncommutative manifold. Note that the usual geometry

of a commutative manifold $\mathcal{M}$ is fully described by using its corresponding algebra of smooth functions $(\mathcal{C}^\infty(\mathcal{M}), \cdot) =: \mathcal{A}$. This algebra $\mathcal{A}$ is covariant under the Hopf algebra $H = (\mathcal{U}\Xi, m, \Delta, \epsilon, S)$. Namely, we have

$$
\begin{aligned}
v(fg) = \mathcal{L}_v(fg) &= \mathcal{L}_v(f)g + f\mathcal{L}_v(g) \\
&= v(f)g + fv(g) \\
&= m[\Delta(v)(f \otimes g)] \qquad \forall f, g \in \mathcal{A} \text{ and } v \in \Xi(\mathcal{M}).
\end{aligned}
\tag{16}
$$

However, the algebra $\mathcal{A}$ is commutative and fails to be covariant under the Hopf algebra $H_\mathcal{F}$ due to $\Delta_\mathcal{F}$, which now encodes a deformed Leibniz rule. Luckily, the algebra $\mathcal{A}$ can be "fixed" by changing the product, i.e., by promoting it into a noncommutative algebra $\mathcal{A}_\star = (\mathcal{C}^\infty(\mathcal{M}), \star)$ so that it becomes covariant under $H_\mathcal{F}$. The $\star$ is called the star-product and is fully determined by the Drinfeld twist $\mathcal{F}$

$$
f \star g := m\left(\mathcal{F}^{-1} f \otimes g\right) =: m_\star(f \otimes g), \quad \forall f, g \in \mathcal{C}^\infty(\mathcal{M}).
\tag{17}
$$

The algebra $\mathcal{A}_\star$ is, in general, noncommutative, since $f \star g \neq g \star f$, but, due to the cocycle condition for the twist (12), it is associative:

$$
f \star (g \star h) = (f \star g) \star h, \quad \forall f, g, h \in \mathcal{C}^\infty(\mathcal{M}).
\tag{18}
$$

In addition, the covariance is now given by

$$
v(f \star g) = m_\star[\Delta_\mathcal{F}(v)(f \otimes g)].
\tag{19}
$$

Therefore, it is often said that the Hopf algebra $H_\mathcal{F}$ describes the symmetries of a noncommutative manifold underlying the algebra $\mathcal{A}_\star = (\mathcal{C}^\infty(\mathcal{M}), \star)$.

The twist $\mathcal{F}$ can be used to construct the $\star$-Lie algebra of vector fields $\Xi_\star$ defined by deforming the Lie bracket (2)

$$
[\cdot, \cdot] \longrightarrow [\cdot, \cdot]_\star := [\cdot, \cdot]\mathcal{F}^{-1} = [\bar{f}^A(\cdot), \bar{f}_A(\cdot)]
\tag{20}
$$

where $\mathcal{F}^{-1} = \bar{f}^A \otimes \bar{f}_A$ [33–36]. Explicitly, for two vector fields, we have

$$
[u, v]_\star = [\bar{f}^A(u), \bar{f}_A(v)] = u \star v - \bar{R}^A(u)\bar{R}_A(v)
\tag{21}
$$

where $\mathcal{R}^{-1} = \bar{R}^A \otimes \bar{R}_A = \mathcal{F}\mathcal{F}_{21}^{-1}$ is the inverse $R$-matrix (the $R$-matrix is defined as $\mathcal{R}(h \star g) = g \star h$) and $\mathcal{F}_{21}^{-1} = \bar{f}_A \otimes \bar{f}^A$. In the $\star$-Lie algebra, we have the $\star$-Jacobi identity

$$
[u, [v, z]_\star]_\star = [[u, v]_\star, z]_\star + [\bar{R}^A(v), [\bar{R}_A(u), z]_\star]_\star.
\tag{22}
$$

The deformed Lie derivative $\mathcal{L}^\star$ is given by

$$
\mathcal{L}_u^\star(v) := [u, v]_\star = \mathcal{L}_{\bar{f}^A(u)}\bar{f}_A(v)
\tag{23}
$$

and satisfies

$$
\mathcal{L}_v^\star \mathcal{L}_w^\star - \mathcal{L}_{\bar{R}^A(w)}^\star \mathcal{L}_{\bar{R}_A(v)}^\star = \mathcal{L}_{[v,w]_\star}^\star, \quad \forall v, w \in \Xi(\mathcal{M}
\tag{24}
$$

and a deformed Leibniz rule compatible with (22)

$$
\mathcal{L}_v^\star(T \otimes_\star T') = \mathcal{L}_v^\star(T) \otimes_\star T' + \bar{R}^A(T) \otimes_\star \mathcal{L}_{\bar{R}_A(v)}^\star(T')
\tag{25}
$$

where $T \otimes_\star T' = \mathcal{F}^{-1}(T \otimes T') = \bar{f}^A(T) \otimes \bar{f}_A(T')$ is the deformed tensor product.

### 2.3. Twisting the Algebra of Exterior Forms

One-forms $\Omega^1(\mathcal{M})$ are dual to vector fields $\Xi(\mathcal{M})$ via the bilinear mapping $\langle \cdot, \cdot \rangle$: $\Xi(\mathcal{M}) \times \Omega^1(\mathcal{M}) \longrightarrow \mathcal{A}$, where, in the local basis of vector fields $\{\partial_\mu\}$ and one-forms $\{dx^\mu\}$, one has

$$\langle v, \omega \rangle = v^\mu \omega_\mu \tag{26}$$

where $v = v^\mu \partial_\mu \in \Xi(\mathcal{M})$ and $\omega = \omega_\mu dx^\mu \in \Omega^1(\mathcal{M})$. Exterior forms (constituted of $p$-forms) $\Omega = \oplus_p \Omega^p(\mathcal{M})$ form an algebra with the wedge product $\wedge : \Omega \times \Omega \longrightarrow \Omega$.

Once again, the twist $\mathcal{F}$ can be used to construct the NC one-forms $\Omega^1_\star$ as dual to vector fields (note that $\Xi(\mathcal{M})$ and $\Xi_\star$ are isomorphic as vector spaces, as well as $\Omega^1(\mathcal{M})$ and $\Omega^1_\star$) $\Xi_\star$ via the $\star$-bilinear mapping $\langle \cdot, \cdot \rangle_\star : \Xi_\star \times \Omega^1_\star \longrightarrow \mathcal{A}_\star$, where

$$\langle v, \omega \rangle_\star := \langle \cdot, \cdot \rangle \mathcal{F}^{-1}(v \otimes \omega) = \left\langle \bar{f}^A(v), \bar{f}_A(\omega) \right\rangle \tag{27}$$

and satisfies the $\mathcal{A}_\star$-linearity properties

$$\langle f \star v, \omega \star g \rangle_\star = f \star \langle v, \omega \rangle_\star \star g, \quad \langle v, f \star \omega \rangle_\star = \bar{R}^A(f) \star \langle \bar{R}_A(v), \omega \rangle_\star \quad \forall f, g \in \mathcal{C}^\infty(\mathcal{M}). \tag{28}$$

### 2.4. Twisting the Geometry and NC Gravity

The formalism of Hopf algebras and twists is suitable for further deforming the geometric objects, such as connection $\nabla : \Xi(\mathcal{M}) \longrightarrow \Omega^1(\mathcal{M}) \times \Xi(\mathcal{M})$, covariant derivative $\nabla_u(v) := \langle u, \nabla v \rangle$, Riemann curvature tensor $R : \Xi(\mathcal{M}) \times \Xi(\mathcal{M}) \times \Xi(\mathcal{M}) \longrightarrow \Xi(\mathcal{M})$ and torsion $T : \Xi(\mathcal{M}) \times \Xi(\mathcal{M}) \longrightarrow \Xi(\mathcal{M})$. Here, we immediately give the definitions of $\star$-objects (for more details, see [33,34]).

A $\star$-connection $\nabla^\star : \Xi_\star \longrightarrow \Omega^1_\star \times \Xi_\star$ is a linear mapping satisfying the following Leibniz rule:

$$\nabla^\star(f \star v) = df \otimes_\star v + f \star \nabla^\star(v) \tag{29}$$

There exists an associated covariant derivative $\nabla^\star_u(v) := \langle u, \nabla^\star(v) \rangle_\star$ that, due to (28) and (29), satisfies

$$\nabla^\star_{u+v} w = \nabla^\star_u w + \nabla^\star_v w \tag{30}$$

$$\nabla^\star_{f \star u} w = f \star \nabla^\star_u w \tag{31}$$

$$\nabla^\star_u(f \star v) = \mathcal{L}^\star_u(f) \star v + \bar{R}^A(f) \star \nabla^\star_{\bar{R}_A(u)} v \tag{32}$$

Note that the $\star$-covariant derivative $\nabla^\star_u(f)$ evaluated on a function is equal to the action of a $\star$-Lie derivative $\mathcal{L}^\star_u(f)$ as in the commutative case. Equations (30)–(32) are often considered as the "axioms" for defining the $\star$-covariant derivative. The $\star$-curvature $R^\star$ and $\star$-torsion $T^\star$ associated to a connection $\nabla^\star$ are defined as

$$R^\star(u, v, w) := \nabla^\star_u \nabla^\star_v w - \nabla^\star_{\bar{R}^A(v)} \nabla^\star_{\bar{R}_A(u)} w - \nabla^\star_{[u,v]_\star} w \tag{33}$$

$$T^\star(u, v) := \nabla^\star_u v - \nabla^\star_{\bar{R}^A(v)} \bar{R}_A(u) - [u, v]_\star \tag{34}$$

Note that the definitions of $R^\star$ and $T^\star$ are analogous to the commutative ones up to the inclusion of the $R$-matrix. Namely, since the noncommutativity of the $\star$-product is regulated by the $R$-matrix whenever we have to permute the order of the elements in the commutative definitions, one needs to include the $R$-matrix contribution (see [36] for more details).

If we assume that there exist a local basis of vector fields $\{e_i\}$ and a dual basis of one-forms $\{\theta^i\}$ such that

$$\left\langle e_i, \theta^j \right\rangle_\star = \delta^j_i \tag{35}$$

we can define the coefficients $R_{ijk}^{\star}{}^{l}$ and $T_{ij}^{\star}{}^{l}$ as

$$R_{ijk}^{\star}{}^{l} = \left\langle R^{\star}(e_i, e_j, e_k), \theta^l \right\rangle_{\star}, \quad T_{ij}^{\star}{}^{l} = \left\langle T^{\star}(e_i, e_j), \theta^l \right\rangle_{\star} \tag{36}$$

and the $\star$-Ricci curvature tensor $\mathrm{R}_{ij}^{\star}$ is given by

$$\mathrm{R}_{ij}^{\star} = \left\langle \theta^l, R^{\star}(e_l, e_i, e_j) \right\rangle_{\star} \tag{37}$$

At this point, we are ready to write the NC version of the vacuum Einstein equation as

$$\mathrm{R}_{ij}^{\star} = 0. \tag{38}$$

Note that all our algebraic $(\mathcal{A}_{\star}, \Xi_{\star}, \Omega_{\star}, H_{\mathcal{F}})$ and geometric objects $(\nabla^{\star}, R^{\star}, T^{\star})$ reduce to the usual ones $(\mathcal{C}^{\infty}(\mathcal{M}), \Xi(\mathcal{M}), \Omega(\mathcal{M}), H, \nabla, R, T)$ in the commutative limit. The inclusion of full $\star$-Riemannian geometry will require the introduction of a metric tensor $g$ and the notion of a torsion-free and metric-compatible connection $\Gamma$. Both of these can be defined in general (see [33,34]), which enables one to define the $\star$-Ricci scalar R and the full Einstein equation

$$\mathrm{R}_{ij}^{\star} - \frac{1}{2} g_{ij} \star \mathrm{R}^{\star} + g_{ij}\Lambda = 8\pi \mathrm{T}_{ij}^{\star}, \tag{39}$$

where $\mathrm{T}_{ij}^{\star}$ is the noncommutative energy–momentum tensor and $\Lambda$ is the cosmological constant. This defines our NC gravity. We will skip further discussion on the general $\star$-formalism and, in the forthcoming subsections, focus on concrete examples.

### 2.5. Angular Twist

An angular twist is a special example of an abelian Drinfeld twist defined by (here, we consider that the twist, or, more specifically, the vector fields $\partial_t$ and $\partial_{\phi}$, always act as Lie derivatives)

$$\mathcal{F} = e^{-\frac{ia}{2}(\partial_t \otimes \partial_{\phi} - \partial_{\phi} \otimes \partial_t)} = 1 \otimes 1 - \frac{ia}{2}(\partial_t \otimes \partial_{\phi} - \partial_{\phi} \otimes \partial_t) + \mathcal{O}(a^2). \tag{40}$$

The twist defined in (40) is abelian since $[\partial_t, \partial_{\phi}] = 0$ and Drinfeld because it satisfies (12). It was extensively studied in [58–61], where it was used to calculate NC corrections to the entropy of Reissner–Nordström BH [25] and field theory [57].

As described in previous subsections, the angular twist (40) defines the noncommutative algebra $\mathcal{A}_{\star} = (\mathcal{C}^{\infty}(\mathcal{M}), \star)$, where the $\star$-product is given by

$$f \star g = m\left(\mathcal{F}^{-1} f \otimes g\right) = fg + \frac{ia}{2}\left(\frac{\partial f}{\partial t}\frac{\partial g}{\partial \phi} - \frac{\partial f}{\partial \phi}\frac{\partial g}{\partial t}\right) + \mathcal{O}(a^2). \tag{41}$$

Therefore the algebra $\mathcal{A}_{\star}$ is a $\kappa$-deformed (it is called $\kappa$-deformed because the algebra (43) can be written in a general $\kappa$-deformed form

$$[x_{\mu} \overset{\star}{,} x_{\nu}] = \frac{i}{\kappa}\left(u_{\mu}x_{\nu} - u_{\nu}x_{\mu}\right) \tag{42}$$

where $\kappa$ is the deformation parameter and $u_{\mu}$ is a constant unit vector. This Lie-algebra-type noncommutative space (42) can be equipped with a $\kappa$-igl Hopf algebra as its symmetry algebra [73,74]. Note that the most famous example of (42) is the so-called $\kappa$-Minkowski algebra (for which, $u_{\mu} = (1, \vec{0})$) together with its $\kappa$-Poincare–Hopf algebra as its symmetry [52–54], and this explains the origin of the name "$\kappa$-deformed") spacetime, where the commutation relations for coordinates are given by

$$[t \overset{\star}{,} x] = -iay, \quad [t \overset{\star}{,} y] = iax, \quad [x \overset{\star}{,} y] = 0, \tag{43}$$

where $[a \overset{\star}{,} b] := a \star b - b \star a$. In the polar coordinates, we have

$$[r \overset{\star}{,} t] = 0, \quad [r \overset{\star}{,} e^{i\phi}] = 0, \quad [e^{i\phi} \overset{\star}{,} t] = ae^{i\phi}, \tag{44}$$

which is connected to the $\kappa$-cylinder algebra [75].

The angular twist (40) is also a Moyal twist since it can be written like

$$\mathcal{F} = e^{\Theta^{AB}\partial_A \otimes \partial_B}, \tag{45}$$

where $A, B = \{t, r, \phi\}$ and $\Theta^{t\phi} = -\Theta^{\phi t} = -\frac{ia}{2}$ are the only non-zero elements of the constant deformation matrix $\Theta^{AB}$. In the case of a Moyal twist, we have a rather big simplification of the general formulation of the NC gravity. Namely, due to the trivial action of the twist on the basis $\{\partial_\mu\}$ and $\{dx^\alpha\}$, note that

$$\langle \partial_\mu, dx^\nu \rangle_\star = \delta_\mu^\nu \tag{46}$$

and that the $\star$-covariant derivative is determined by

$$\nabla^\star_{\partial_\mu} \partial_\nu := \nabla^\star_\mu \partial_\nu = \Sigma_{\mu\nu}{}^\rho \star \partial_\rho = \Sigma_{\mu\nu}{}^\rho \partial_\rho, \quad \Sigma_{\mu\nu}{}^\rho \in \mathcal{C}^\infty(\mathcal{M}) \tag{47}$$

which leads to

$$R^\star_{\mu\nu\rho}{}^\sigma = \partial_\mu \Sigma_{\nu\rho}{}^\sigma - \partial_\nu \Sigma_{\mu\rho}{}^\sigma + \Sigma_{\nu\rho}{}^\tau \star \Sigma_{\mu\tau}{}^\sigma - \Sigma_{\mu\rho}{}^\tau \star \Sigma_{\nu\tau}{}^\sigma \tag{48}$$

and

$$T^\star_{\mu\nu}{}^\rho = \Sigma_{\mu\nu}{}^\rho - \Sigma_{\nu\mu}{}^\rho. \tag{49}$$

The metric tensor $g$ is given by

$$g = g_{\mu\nu} \star dx^\mu \otimes_\star dx^\nu = g_{\mu\nu} dx^\mu \otimes dx^\nu \tag{50}$$

and it remains undeformed. One can show [33,34] that there exist a unique $\star$-torsion free

$$T^\star_{\mu\nu}{}^\rho = 0 \tag{51}$$

and metric-compatible

$$\nabla^\star_\mu g = 0 \tag{52}$$

$\star$ Levi–Civita connection that is explicitly given by

$$\Sigma_{\mu\nu}{}^\rho = \frac{1}{2} g^{\star\rho\sigma} \star \left( \partial_\mu g_{\nu\sigma} + \partial_\nu g_{\mu\sigma} - \partial_\sigma g_{\mu\nu} \right) \tag{53}$$

where $g^{\star\rho\sigma}$ is the unique $\star$-inverse satisfying

$$g^{\star\alpha\rho} \star g_{\rho\beta} = \delta_\beta^\alpha, \quad g_{\mu\rho} \star g^{\star\rho\nu} = \delta_\mu^\nu \tag{54}$$

and is explicitly given as

$$g^{\star\alpha\beta} = g^{\alpha\beta} - g^{\gamma\beta} \Theta^{AB} (\partial_A g^{\alpha\sigma})(\partial_B g_{\sigma\gamma}) + \mathcal{O}(a^2) \tag{55}$$

where $g^{\alpha\beta}$ is the usual inverse

$$g^{\alpha\sigma} g_{\sigma\beta} = \delta_\beta^\alpha. \tag{56}$$

Now, it is easy to see that, if the metric $g$ has Killing vectors $K_\alpha$ compatible with the Moyal twist, i.e., $[K_\alpha, \partial_\beta] = 0$, the $\star$-curvature and $\star$-Levi–Civita (53) are undeformed and, in that case, any $g$ that satisfies the commutative Einstein equation will also satisfy the $\star$-Einstein Equation (39) provided that the energy–momentum tensor inherits the symmetry,

i.e., $\mathcal{L}_{K_\alpha}(\mathrm{T}) = 0$. However, the NC geodesic motion will change and so will the equation of motion for the metric perturbations (see [33,34]).

### 2.6. Noncommutative QBTZ

Using the formalism outlined so far, we want to deform a particular manifold described by the QBTZ metric [62–65] with the angular twist (40). The QBTZ metric is given by

$$ds^2 = -f dt^2 + \frac{1}{f} dr^2 + r^2 d\phi^2, \quad f(r) = -M + \frac{r^2}{l^2} - 2Q^2 \ln\left(\frac{r}{l}\right) \tag{57}$$

and, together with the electromagnetic potential $A = A_\mu dx^\mu$

$$A_\mu = \left(-Q \ln\left(\frac{r}{l}\right), \vec{0}\right) \tag{58}$$

satisfies the coupled Einstein–Maxwell equations (coupled Einstein–Maxwell equations consist of the Einstein equation with the energy–momentum tensor being the Maxwell stress tensor and the Maxwell equations in the curved background $g$). Here, $M$ and $Q$ are the mass and charge of the black hole, while $1/l^2$ is the cosmological constant rendering the QBTZ-metric asymptotically $AdS$. The QBTZ has time-translation and azimuthal symmetry since

$$\mathcal{L}_{\partial_t}(g) = \mathcal{L}_{\partial_\phi}(g) = \mathcal{L}_{\partial_t}(A) = \mathcal{L}_{\partial_\phi}(A) = 0 \tag{59}$$

implying that $\partial_t$ and $\partial_\phi$ are its Killing vector fields. Whenever we want to deform a manifold that has time-translation and azimuthal symmetry (rotation in the $x - y$ plane)—that is, whenever the metric $g$ has $\partial_t$ and $\partial_\phi$ as Killing vectors—the angular twist (40) becomes an abelian affine Killing twist [33]. One can easily see that deforming the QBTZ with the angular twist will lead to the conclusion that the whole NC differential geometry is undeformed and that QBTZ is also a solution of the $\star$-Einstein Equation (39) with $\mathrm{T}^\star = \mathrm{T} = \mathrm{T}_{\mu\nu} dx^\mu \otimes dx^\mu$, where $\mathrm{T}_{\mu\nu}$ are the components of the Maxwell stress tensor. In order to see this more explicitly, it is enough to calculate

$$\nabla_\mu^\star \partial_\nu = \Sigma_{\mu\nu}{}^\rho \star \partial_\rho = \Gamma_{\mu\nu}{}^\rho \partial_\rho \tag{60}$$

where $\Sigma_{\mu\nu}{}^\rho = \Gamma_{\mu\nu}{}^\rho = \frac{1}{2} g^{\rho\sigma} \left(\partial_\mu g_{\nu\sigma} + \partial_\nu g_{\mu\sigma} - \partial_\sigma g_{\mu\nu}\right)$ are the undeformed Levi–Civita coefficients leading ultimately to $R^\star = R$, etc.

Therefore, we can consider the QBTZ to be a fixed commutative and NC background at the same time, i.e., $QBTZ = NCQBTZ$. This is very convenient because we effectively have that the spacetime part, i.e., the geometry, can be treated completely classically. However, when considering NC field theory—for example, if we introduce a charged scalar probe $\hat{\Phi}$ into the scenario—the scenario become interesting. As shown in [58–60], there will be a nontrivial NC correction to the coupling between the $U(1)$ field and the charged scalar field. We will look at this in more detail in the next subsection.

### 2.7. NC Klein–Gordon Equation in $(NC)QBTZ$ Background

Since we showed that the NCQBTZ, which is the QBTZ deformed with an angular twist (40), can be treated as a classical curved background, the action for an NC-charged scalar field is given by

$$S[\Phi] = \int d^4x \sqrt{-g} \left(g^{\mu\nu} (D_\mu \hat{\Phi})^+ \star D_\nu \hat{\Phi} - \frac{\mu^2}{\hbar^2} \hat{\Phi}^+ \star \hat{\Phi}\right), \tag{61}$$

where $D_\mu$ is the $U(1)_\star$-covariant derivative given by

$$D_\mu \hat{\Phi} = \partial_\mu \hat{\Phi} - i\frac{q}{\hbar} \hat{A}_\mu \star \hat{\Phi} \tag{62}$$

and where $q$ and $\mu$ are the charge and mass of the scalar field $\hat{\Phi}$. Such a defined action (61) is invariant under infinitesimal $U(1)_\star$ gauge transformations

$$\delta_{\hat{\Lambda}}\hat{\Phi} = i\hat{\Lambda}\star\hat{\Phi}, \quad \delta_{\hat{\Lambda}}\hat{A}_\mu = \partial_\mu\hat{\Lambda} + i(\hat{\Lambda}\star\hat{A}_\mu - \hat{A}_\mu\star\hat{\Lambda}), \quad \delta_{\hat{\Lambda}}g_{\mu\nu} = 0 \tag{63}$$

where $\hat{\Lambda}$ is the NC gauge parameter. In order to express the NC field $\hat{\Phi}$ and $\hat{A}_\mu$ in terms of their commutative counterparts $\Phi$ and $A_\mu$, one needs to use the so-called Seiberg–Witten (SW) map [76,77]. The SW map for an abelian twist is known up to all orders in the deformation parameter $a$ [78] and is explicitly given as

$$\hat{\Phi} = \Phi - \frac{q}{4\hbar}\Theta^{\rho\sigma}A_\rho(\partial_\sigma\Phi + (\partial_\sigma - i\frac{q}{\hbar}A_\sigma)\Phi) + \mathcal{O}(a^2), \quad \hat{A}_\mu = A_\mu - \frac{q}{2\hbar}\Theta^{\rho\sigma}A_\rho(\partial_\sigma A_\mu + F_{\sigma\mu}) + \mathcal{O}(a^2) \tag{64}$$

where $F_{\mu\nu} = \partial_\mu A_\nu - \partial_\nu A_\mu$ are the components of the electromagnetic field-strength two-form. Using (40) and (64), the action (61) up to the first order in $a$ is given by

$$S = \int d^4x\sqrt{-g}\left[g^{\mu\nu}(D_\mu\Phi)^+D_\nu\Phi - \frac{\mu^2}{\hbar^2}\Phi^+\Phi + \mathcal{L}(a)\right] + \mathcal{O}(a^2) \tag{65}$$

where $\mathcal{L}(a)$ is the NC correction

$$\mathcal{L}(a) = \frac{\mu^2q^2}{2\hbar^2}\Theta^{\alpha\beta}F_{\alpha\beta}\Phi^+\Phi + \frac{q^2\Theta^{\alpha\beta}}{2}g^{\mu\nu}\left(-\frac{1}{2}(D_\mu\Phi)^+F_{\alpha\beta}D_\nu\Phi + (D_\mu\Phi)^+F_{\alpha\nu}D_\beta\Phi + (D_\beta\Phi)^+F_{\alpha\mu}D_\nu\Phi\right) \tag{66}$$

Now, we vary the action (65) with respect to $\Phi^+$ to obtain the equation of motion for $\Phi$ and, after using the ansatz $\Phi = e^{-\frac{i}{\hbar}Et}R(r)e^{im\phi}$, we obtain the radial equation of motion

$$R_m'' + \frac{1}{f}\left[\frac{1}{f}\frac{(E - qQ\ln(\frac{r}{l}))^2}{\hbar^2} - \frac{m^2}{r^2} - \frac{\mu^2}{\hbar^2}\right]R_m + \frac{2}{rf}\left(\frac{r^2}{l^2} - Q^2 + \frac{f}{2}\right)R_m' + im\frac{aqQ}{\hbar r^2 f}\left[rf\frac{d}{dr} + r\left(\frac{r}{l^2} - \frac{Q^2}{r^2}\right)\right]R_m = 0 \tag{67}$$

where $E$ and $m$ are separation constants corresponding to the energy and angular momentum (magnetic quantum number), and we explicitly used the QBTZ-metric (57). Radial Equation (67) is the central result of this paper and, in the following sections, we use it to calculate the NC correction to the entropy of QBTZ via the brick-wall method.

## 3. Black Hole Entropy in the Brick Wall Model

In order to calculate the entropy of a BH in general, it is crucial to regularize the UV divergence and isolate the relevant physical contribution [19,79]. One of the simplest ways to achieve this was presented a long time ago by 't Hooft [6,7] using the so-called *brick wall* method. Alongside the brick wall method, the BH entropy can be described using the entanglement of the degrees of freedom between the two sides of the horizon [80,81] and by using the Wald entropy formula within the effective field theory approach to quantum gravity [21,22,82]. It has been shown [19,83] that all three approaches are related due to the fact that all of them have an almost identical UV divergence of the BH entropy.

The brick wall method is a semi-classical approach, where the BH is considered as a fixed background that is in thermal equilibrium with a thermal bath of some surrounding quantum matter fields at the Hawking temperature. Therefore, the canonical entropy of the matter field outside the BH horizon is related to the entropy of BH itself. When calculating the canonical entropy, the crucial ingredient is the density of states. The density of states diverges at the horizon and this is the reason why one uses a cutoff $h$, i.e., the brick wall, imposing that, at this point $r_+ + h$, the matter fields outside the BH horizon vanish. The value of the cutoff is determined from the matching condition, where the leading divergent part of the canonical entropy obeys the Bekenstein–Hawking area law. The canonical entropy is given by

$$S = \beta^2\frac{dF}{d\beta} \tag{68}$$

where $F$ is the free energy of the matter field at the inverse temperature $\beta$ and is given by [6,7]

$$F = -\int_0^\infty \frac{N(E)}{e^{\beta E} - 1} dE. \tag{69}$$

In order to evaluate (69), the key ingredients are the energy $E$ (the spectrum of the field) and the density of states $N(E)$, i.e., the number of eigenmodes of the matter field. Both of them are determined by solving the equation of motion for a surrounding field in the fixed BH background and imposing the brick wall boundary conditions, i.e., that the fields are vanishing close to the horizon and in the spatial infinity. The first condition will regularize the UV infinities, whereas the second regularizes the IR infinities, which will occur in the density of states due to the contribution coming from the vacuum surrounding the system at large distances, but this can be omitted [6]. For scalar fields, everything is governed by the radial part of the Klein–Gordon equation and one uses the WKB method to solve it. Now, we move on to our concrete example of the NC scalar field on an NCQBTZ background.

### 3.1. WKB and the Density of States $N(E)$

The radial part of the equation of motion for the NC scalar field on an NCQBTZ background is given in (67) and, since we are unable to solve it analytically, we used the WKB method. To carry this out, we first used an ansatz

$$R_m = \frac{\psi_m}{\sqrt{r}\sqrt{f}} \tag{70}$$

so that (67) can be put in the following form (note that Equation (71) slightly differs from the corresponding one in [84] due to the appearance of the $A$, $B$ and $C$ term. They appear here because of two reasons. One is because we are looking at a Klein–Gordon equation for a charged scalar and the other is because of the NC corrections. When $q$ and $a$ tend to zero, we recover the equation in [84]):

$$\psi_m'' + \left(\frac{A(r)}{\hbar} + B(r)\right)\psi_m' + \left(\frac{V(r)^2}{\hbar^2} + \frac{C(r)}{\hbar} - \Delta(r)\right)\psi_m = 0 \, . \tag{71}$$

where

$$V^2 = \frac{-\frac{\hbar^2 m^2 f(r)}{r^2} + \left(E - Qq \ln\left(\frac{r}{l}\right)\right)^2}{f^2(r)} \tag{72}$$

$$\Delta = -\frac{Q^2 \frac{d}{dr} f(r)}{r f^2(r)} - \frac{Q^2}{r^2 f(r)} + \frac{\frac{d^2}{dr^2} f(r)}{2 f(r)} - \frac{3\left(\frac{d}{dr} f(r)\right)^2}{4 f^2(r)} - \frac{1}{4r^2} + \frac{r \frac{d}{dr} f(r)}{l^2 f^2(r)} + \frac{1}{l^2 f(r)} \tag{73}$$

$$A = \frac{iQamq}{r} \tag{74}$$

$$C = \frac{iQamq\left(-2Q^2 l^2 - l^2 r^2 \frac{d}{dr} f(r) - l^2 r f(r) + 2r^3\right)}{2l^2 r^3 f(r)} \tag{75}$$

$$B = -\frac{2Q^2}{r f(r)} - \frac{\frac{d}{dr} f(r)}{f(r)} + \frac{2r}{l^2 f(r)} \tag{76}$$

The $V^2$ function is related to the effective potential (for the $\mu = 0$ case) together with the energy, $A$ and $C$ are the NC correction to the $\psi'$ and $\psi$ term, respectively, $B$ is the correction to the $\psi'$ term due to charge $q$ and $\Delta$ is the standard term related to the existence of the term $R'$ in (67).

In order to solve (70), we used the WKB-ansatz

$$\psi_m = \frac{c_0}{\sqrt{P(r)}} \, e^{\frac{i}{\hbar} \int^r P(r')dr'} \tag{77}$$

to obtain a differential equation for $P(r)$:

$$\frac{1}{\hbar^2}\left(V^2 - P^2 + iPA\right) + \frac{1}{\hbar}\left(C + iPB - \frac{P'}{2P}A\right) = \frac{P''}{2P} - \frac{3}{4}\frac{P'^2}{P^2} + \Delta + \frac{P'}{2P}B. \tag{78}$$

We are looking for a solution that is a power series in $\hbar$, namely

$$P(r) = \sum_{n=0}^{\infty} \hbar^n P_n(r). \tag{79}$$

Substituting the series (79) into (78), we obtain, for the $P_n$ up to $n = 2$,

$$P_0 = i\frac{A}{2} \pm \sqrt{V^2 - \frac{A^2}{4}} \tag{80}$$

$$P_1 = \frac{A\frac{P_0'}{2P_0} - iBP_0 - C}{iA - 2P_0} \tag{81}$$

$$P_2 = \frac{P_1^2 - iP_1B + A\frac{P_1'}{2P_0} - A\frac{P_0'P_1}{2P_0^2} - \frac{3}{4}\frac{P_0'^2}{P_0} + \frac{P_0''}{2P_0} + B\frac{P_0'}{2P_0} + \Delta}{iA - iP_0} \tag{82}$$

From now on, we will focus on the lowest order in the WKB approximation, i.e., on $P_0$. Using (72) and (80) and choosing the $+$ sign, we obtain

$$P_0 = -\frac{Qamq}{2r} + \sqrt{\frac{Q^2a^2m^2q^2}{4r^2} + \frac{-\frac{\hbar^2m^2f(r)}{r^2} + \left(E - Qq\ln\left(\frac{r}{l}\right)\right)^2}{f^2(r)}}. \tag{83}$$

Since we expect that the NC scale governed by the deformation parameter $a$ is small compared to the other scales in the problem, i.e., it is comparable with the Planck scale, we further expanded $P_0$ up to $a^2$ (we expanded up to $a^2$ because, later, we will see that the linear terms do not contribute to the entropy $S$):

$$P_0 = \frac{Q^2a^2m^2q^2f(r)}{8r^2\sqrt{-\frac{\hbar^2m^2f(r)}{r^2} + \left(E - Qq\ln\left(\frac{r}{l}\right)\right)^2}} - \frac{Qamq}{2r} + \frac{\sqrt{-\frac{\hbar^2m^2f(r)}{r^2} + \left(E - Qq\ln\left(\frac{r}{l}\right)\right)^2}}{f(r)}. \tag{84}$$

The density of states $N(E)$ is defined by [6,7]:

$$N(E) = \sum_{n=0}^{\infty} N_n(E) = \frac{1}{\pi\hbar} \int_{r_++h}^{L} dr \int_{-m_{max}}^{m_{max}} dm \; P(r) \tag{85}$$

where $r_+$ is the outer horizon, $h$ is the brick wall cutoff, $L$ is the IR cutoff and $m_{max}$ is the maximal value of the magnetic quantum number $m$ such that $P$ remains real and is given by

$$m_{max} = \frac{r}{\hbar\sqrt{f}}\left(E - qQ\ln\left(\frac{r}{l}\right)\right). \tag{86}$$

In the lowest order in WKB, we have

$$N_0 = \frac{1}{\hbar\pi} \int_{r_+ + h}^{L} dr \int_{-m_{\max}}^{m_{\max}} P_0 dm$$

$$= \frac{1}{\hbar^2 \pi} \int_{r_+ + h}^{L} \frac{r}{\sqrt{f}} \left[ \frac{1}{f} \int_0^{\mathcal{E}} d\Lambda \frac{\mathcal{G}}{\sqrt{\Lambda}} + \frac{a^2 q^2 Q^2}{8\hbar^2} \int_0^{\mathcal{E}} d\Lambda \frac{\sqrt{\Lambda}}{\mathcal{G}} \right], \tag{87}$$

where

$$\mathcal{G}(\Lambda, \mathcal{E}) = \sqrt{\mathcal{E} - \Lambda}, \quad \mathcal{E} = \left( E - Qq \ln\left(\frac{r}{l}\right) \right)^2, \quad \Lambda = \frac{\hbar^2 m^2 f(r)}{r^2} \tag{88}$$

and the linear term in $a$ vanishes due to $\int_{-m_{\max}}^{m_{\max}} m \, dm = 0$. Next, we evaluated the $\Lambda$ integration and, using

$$\int_0^{\mathcal{E}} \frac{\Lambda}{\mathcal{G}\sqrt{\Lambda}} d\Lambda = \frac{\pi}{2}\mathcal{E}, \quad \int_0^{\mathcal{E}} \frac{\mathcal{G}}{\sqrt{\Lambda}} d\Lambda = \frac{\pi}{2}\mathcal{E} \tag{89}$$

we obtained the density of states $N(E)$ in the lowest WKB order

$$N_0 = \frac{1}{2\hbar^2} \int_{r_+ + h}^{L} dr \left[ E^2 - 2EQq \ln\left(\frac{r}{l}\right) \right] \frac{r}{\sqrt{f}} \left( \frac{1}{f} + \frac{a^2 q^2 Q^2}{8\hbar^2} \right) \tag{90}$$

and we postponed the $r$-integration for later.

### 3.2. Entropy of NCQBTZ

Once we have the density of states $N_0$, using (68) and (69), we can obtain the free energy $F_0$ and canonical entropy $S_0$. Here, we also used the notation

$$F = \sum_{n=0}^{\infty} F_n, \quad S = \sum_{n=0}^{\infty} S_n. \tag{91}$$

After the $E$ integration, we obtain

$$S_0 = \frac{1}{2\hbar^2} \int_{r_+ + h}^{L} dr \left( \frac{6\zeta(3)}{\beta^2} - 4Qq \ln\left(\frac{r}{l}\right) \frac{\zeta(2)}{\beta} \right) \frac{r}{\sqrt{f}} \left( \frac{1}{f} + \frac{a^2 q^2 Q^2}{8\hbar^2} \right) \tag{92}$$

where we used the $\zeta$-function regularization and also subtracted the infinite contribution proportional to $\zeta(1)$, which originates from the electrostatic self-energy of the charge $q$ of the scalar particle.

We are interested in the main contribution to the entropy $S_0$ coming from the horizon. In order to extract this contribution, we split the integration over $r$ into two parts:

$$\int_{r_+ + h}^{L} dr(\ldots) = \int_{r_+ + h}^{R} dr(\ldots) + \int_{R}^{L} dr(\ldots) \tag{93}$$

where we introduced an intermediate scale $R > 0$ such that, in the first term we have $h \ll r_+ \ll R$ and, in the second term, $r_+ \ll R \ll L$. The second term is the contribution of the vacuum surrounding the system at large distances and can be omitted [6,7]. The first contribution can be evaluated in the near horizon limit; namely, we used the near horizon coordinate $x = r - r_+$ and $h \ll r_+$ to obtain

$$S_0 = \frac{1}{\sqrt{h}} \sum_{n=0}^{\infty} h^n f_n \tag{94}$$

where the coefficients $f_n$ are given in Appendix A up to $n = 3$, and the inverse temperature $\beta$ is related to the Hawking temperature via surface gravity $\kappa$:

$$\frac{1}{\beta} = \hbar \frac{\kappa}{2\pi} = \hbar \frac{f'(r_+)}{4\pi}. \tag{95}$$

The brick wall cutoff $h$ is determined by imposing that the most divergent part of $S_0$ as $h \longrightarrow 0$ is equal to the Bekenstein–Hawking entropy $S_{\text{BH}} = \frac{A}{4\hbar}$ [6,7] i.e.,

$$(S_0)_{\text{div}} = \frac{f_0}{\sqrt{h}} = S_{\text{BH}} \tag{96}$$

leading to

$$h = \left( \frac{f_0}{S_{\text{BH}}} \right)^2. \tag{97}$$

Now, we can plug the cutoff (97) into the full expression for the entropy (94) to obtain the dependence of the entropy on the area $A$, i.e., the quantum and NC corrections to the Bekenstein–Hawking area law

$$S_0 = S_{\text{BH}} + \sum_{k=0}^{\infty} \mathcal{V}_k \ln^k \left( \frac{A}{l} \right) + \left( \frac{aqQ}{\hbar} \right)^2 \sum_{k=0}^{\infty} \mathcal{W}_k \ln^k \left( \frac{A}{l} \right) \tag{98}$$

where $A = 2\pi r_+$ is the area of the horizon and the function coefficients $\mathcal{V}_k$ and $\mathcal{W}_k$ depend on $A$ and have an expansion in $\hbar$:

$$\mathcal{V}_k = \frac{1}{\hbar} \sum_{n=0}^{\infty} \hbar^n v_{kn}, \quad \mathcal{W}_k = \frac{1}{\hbar} \sum_{n=0}^{\infty} \hbar^n w_{kn} \tag{99}$$

where the coefficients $v_{kn}$ and $w_{kn}$ are given in the Appendix B up to $n = 5$ and $k = 3$. Note that, up to the linear order in $\hbar$, the expression for the entropy (98) is of the general form presented in [84], but, due to the existence of terms $\ln^k \left( \frac{A}{l} \right)$ in higher orders in $\hbar$, this is no longer true. This is due to the fact that the electromagnetic potential (58) in 2+1 dimensions has a logarithmic dependence, rather than $r^{-1}$, which then explains the appearance of higher powers of ln in the expression for the entropy (98).

## 4. Some Limit Cases

The equation for the entropy (98) appears to be very involved, so we will analyze some of its interesting limiting cases.

### 4.1. Limit Case $q \longrightarrow 0$

The limit $q \longrightarrow 0$ means that the charge of the surrounding scalar field $\Phi$ is negligible. In this case, there is no coupling between the electromagnetic potential $A_\mu$ and $\Phi$, and therefore there are no NC corrections to the entropy. However, we obtain quantum corrections to the entropy of QBTZ, i.e.,

$$\lim_{q \to 0} S = S_{\text{BH}} + \sum_{k=0}^{\infty} \lim_{q \to 0} (\mathcal{V}_k) \ln^k \left( \frac{A}{l} \right) = S_{\text{BH}} + \hbar \mathcal{F}_{QBTZ}(A) \ln \left( \frac{A}{l} \right) \tag{100}$$

where we used

$$\lim_{q \to 0} (\mathcal{V}_k) = \lim_{q \to 0} (\mathcal{V}_0) = \hbar \lim_{q \to 0} v_{02} = \hbar \mathcal{F}_{QBTZ}(A) = -\hbar \frac{9\zeta(3)^2 \left( \frac{A}{\pi l^2} - \frac{4\pi Q^2}{A} \right)}{32\pi^5} \tag{101}$$

This is in complete agreement with [84], and (100) represents the entropy of QBTZ in the lowest WKB order.

*4.2. Limit Case $Q \longrightarrow 0$*

The $Q \longrightarrow 0$ limit means that the charge of the BH is negligible. If so, there is no electromagnetic potential $A_\mu$, no coupling with $\Phi$ and no NC corrections. In addition, the QBTZ reduces to the spinless BTZ metric, i.e., for the entropy, we obtain

$$\lim_{Q \to 0} S = S_{BTZ} + \sum_{k=0}^{\infty} \lim_{Q \to 0} (\mathcal{V}_k) \ln^k \left( \frac{A_{BTZ}}{l} \right) = S_{BTZ} + \hbar \mathcal{F}_{BTZ}(A) \ln \left( \frac{A_{BTZ}}{l} \right) \tag{102}$$

where we used

$$\lim_{Q \to 0} (\mathcal{V}_k) = \lim_{Q \to 0} (\mathcal{V}_0) = \hbar \lim_{Q \to 0} v_{02} = \hbar \mathcal{F}_{BTZ}(A) = -\hbar \frac{9\zeta(3)^2 A_{BTZ}}{32\pi^6 l^2} \tag{103}$$

and

$$S_{BTZ} = \frac{A_{BTZ}}{4\hbar}, \quad A_{BTZ} = 2\pi r_{BTZ}, \quad r_{BTZ} = l\sqrt{M}. \tag{104}$$

This is also in complete agreement with [84], and (102) represents the entropy of BTZ in the lowest WKB order.

*4.3. The Almost BTZ Limit*

The angular twist (40) does not deform either the BTZ metric or its coupling to the scalar probe. Hence, taking limits of $Q, q \longrightarrow 0$ in (98) simply reproduces the commutative results. The NC corrections only appear if we have a charged BH (in our case, QBTZ) and a charged scalar probe $\Phi$. Therefore, it is interesting to take a closer look at the QBTZ with a very small charge $Q$ (but not negligible) in order to compare the NC corrections with respect to the (almost) BTZ black hole. We will investigate (98) for small $Q$; that is, we expand everything to the lowest order in the BH charge $Q$. Since the NC correction is proportional to $Q^2$, it is enough to expand the $S_0$ up to quadratic terms in $Q$. To carry this out, we first need to find the expansion of $r_+$. Since $r_+$ is given in terms of Lambert function $W$ [85], in order to avoid the asymptotics and a complex analysis, we solved the condition for the horizon of QBTZ metric $f(r_+) = 0$ perturbatively. First, we wrote (the symbol $\mathcal{O}(\sim)$ represents a function of the form $\alpha_1 \xi + \alpha_2 \xi^2 + \ldots$, where $\alpha_n$ are some dimensional parameters)

$$\begin{aligned} r_+ &= r_{\text{BTZ}} + \tilde{r}, \quad \tilde{r} = \mathcal{O}(Q^2) \\ &\implies r_+^2 = r_{\text{BTZ}}^2 + 2 r_{\text{BTZ}} \tilde{r} + \mathcal{O}(\tilde{r}^2) \\ &\implies \ln \left( \frac{r_+}{l} \right) = \ln \left( \frac{r_{\text{BTZ}}}{l} \right) + \frac{\tilde{r}}{r_{\text{BTZ}}} + \mathcal{O}(\tilde{r}^2) \end{aligned} \tag{105}$$

which gives

$$\tilde{r} = \frac{l^2}{r_{\text{BTZ}}} Q^2 \ln \left( \frac{r_{\text{BTZ}}}{l} \right) + \mathcal{O}(Q^4) \tag{106}$$

and the expansion for $r_+$ is given by

$$r_+ = r_{\text{BTZ}} \left( 1 + \left( \frac{l}{r_{\text{BTZ}}} \right)^2 \ln \left( \frac{r_{\text{BTZ}}}{l} \right) Q^2 \right) \tag{107}$$

Next, we expanded the entropy (98) as a series in $Q$ and obtained

$$S = S_{\text{BTZ}} + \sum_{n=0}^{\infty} s_n Q^n + \frac{a^2 q^2}{\hbar^2} \sum_{n=0}^{\infty} z_n Q^n \tag{108}$$

where $s_n$ and $z_n$ were calculated up to $n = 2$, i.e.,

$$s_0 = -\frac{9 A_{\text{BTZ}} \zeta(3)^2 \hbar}{32 \pi^6 l^2} \tag{109}$$

$$s_1 = \hbar^0 \left( \frac{3\zeta(2)\zeta(3) q \ln\left(\frac{A_{\text{BTZ}}}{l}\right)}{2\pi^4} - \frac{3\zeta(2)\zeta(3) q \ln(2\pi)}{2\pi^4} + \frac{3\zeta(2)\zeta(3) q}{4\pi^4} \right) + \hbar^2 \left( \frac{9\zeta(2)\zeta(3)^3 q}{32\pi^{10} l^2} \right) \tag{110}$$

$$s_2 = \frac{1}{\hbar} \left( \frac{2\zeta(2)^2 l^2 q^2 \ln\left(\frac{A_{\text{BTZ}}}{l}\right)^2}{\pi^2 A_{\text{BTZ}}} - \frac{2\zeta(2)^2 l^2 q^2 \ln\left(\frac{A_{\text{BTZ}}}{l}\right)}{\pi^2 A_{\text{BTZ}}} + \frac{4\zeta(2)^2 l^2 q^2 \ln(2\pi) \ln\left(\frac{A_{\text{BTZ}}}{l}\right)}{\pi^2 A_{\text{BTZ}}} - \frac{2\zeta(2)^2 l^2 q^2 \ln(2\pi)^2}{\pi^2 A_{\text{BTZ}}} + \right.$$

$$\left. + \frac{2\zeta(2)^2 l^2 q^2 \ln(2\pi)}{\pi^2 A_{\text{BTZ}}} + \frac{\pi^2 l^2 \ln\left(\frac{A_{\text{BTZ}}}{l}\right)}{A_{\text{BTZ}}} - \frac{\pi^2 l^2 \ln(2\pi)}{A_{\text{BTZ}}} \right) + \hbar^0 \cdot 0 \tag{111}$$

$$+ \hbar \left( -\frac{9\zeta(2)^2 \zeta(3)^2 q^2 \ln\left(\frac{A_{\text{BTZ}}}{l}\right)}{4\pi^8 A_{\text{BTZ}}} + \frac{9\zeta(2)^2 \zeta(3)^2 q^2 \ln(2\pi)}{4\pi^8 A_{\text{BTZ}}} - \frac{9\zeta(3)^2 \ln\left(\frac{A_{\text{BTZ}}}{l}\right)}{8\pi^4 A_{\text{BTZ}}} + \frac{9\zeta(3)^2}{8\pi^4 A_{\text{BTZ}}} + \frac{9\zeta(3)^2 \ln(2\pi)}{8\pi^4 A_{\text{BTZ}}} \right)$$

and

$$z_0 = z_1 = 0, \quad z_2 = \hbar \left( -\frac{9 A_{\text{BTZ}}^3 \zeta(3)^2}{512 \pi^8 l^4} \right) + \hbar^3 \left( -\frac{27 A_{\text{BTZ}}^3 \zeta(3)^4}{4096 \pi^{14} l^6} \right) \tag{112}$$

## 5. Final Remarks

The brick wall method is one of the most widely used methods for calculating corrections to the Bekenstein–Hawking entropy [84]. It is important to note that even the lowest order in WKB can provide corrections of the same structure as higher orders if one expands the metric beyond the linear order in the horizon [84]. Like in [25], we present a derivation of the NC correction to Bekenstein–Hawking entropy steaming from a Drinfeld twist, which is compatible with the symmetries of the background metric, i.e., we used a Killing twist [33,34]. Contrary to the general form for the entropy corrections found in [25,84], the NC corrections to the entropy of QBTZ exhibit extra terms that are polynomials in logarithms. This peculiarity is present due to the fact that both the electromagnetic potential (58) and QBTZ metric (57) have an explicit logarithmic dependence on $r$ that later propagates in the WKB expansion. It is generally believed that the appearance of logarithmic dependence in the corrections to the entropy of black holes is due to the nonlocality of the quantum nature of gravity. If so, we can conclude that, in the special example of QBTZ deformed via the angular twist (40), the NC corrections to entropy seem to suggest that the quantum aspects of gravity in 2+1 dimensions have a higher degree of nonlocality than the corresponding theory in 3+1 dimensions [25].

It is important to note that the universal nature of the UV divergence of the BH entropy is related to the fact that the von Neumann algebra of observables in QFT in curved backgrounds is of type-III [86–88]. As supported by the results in [25] and in this paper, it seems that, from a perturbative standpoint, these UV divergences persist even in the NC framework, suggesting that noncommutativity does not change the typology of the corresponding von Neumann algebra of observables.

**Author Contributions:** T.J. and F.P. have equally contributed to this research. All authors have read and agreed to the published version of the manuscript.

**Funding:** This research has been supported by Croatian Science Foundation project IP-2020-02-9614.

**Data Availability Statement:** Not applicable.

**Acknowledgments:** The authors would like to thank K.S. Gupta, A. Samsarov, I. Smolić, K. Delić and J. Hammoud for various discussions in the early stages of this work.

**Conflicts of Interest:** The authors declare no conflict of interest.

## Appendix A. Entropy in Terms of the Brick Wall Cutoff $h$

In order to evaluate the contribution coming from the horizon of the entropy $S_0$, we need to integrate (92) in the near horizon limit

$$S = \frac{1}{2\hbar^2} \int_h dx \left[ \frac{6\zeta(3)}{\beta^2} - 4Qq\left(\ln\left(\frac{r_+}{l}\right) + \frac{x}{r_+}\right)\frac{\zeta(2)}{\beta}\right] \frac{x+r_+}{\sqrt{f'(r_+)x}}\left(\frac{1}{f'(r_+)x} + \frac{a^2q^2Q^2}{8\hbar^2}\right). \tag{A1}$$

where the upper bound contribution is omitted, and we also used

$$\ln\left(\frac{r}{l}\right) = \ln\left(\frac{r_+}{l}\right) + \frac{x}{r_+} + \mathcal{O}(x^2), \quad f(r) = f'(r_+)x + \mathcal{O}(x^2) \tag{A2}$$

since $f(r_+) = 0$ and $f'(r_+) = \left(\frac{2r_+}{l^2} - \frac{2Q^2}{r_+}\right)$. After the $x$ integration and using

$$\frac{1}{\beta} = \hbar\frac{\kappa}{2\pi} = \hbar\frac{f'(r_+)}{4\pi}. \tag{A3}$$

we obtained

$$S_0 = \frac{1}{\sqrt{h}} \sum_{n=0}^{\infty} h^n f_n \tag{A4}$$

where, up to $n = 3$,

$$f_0 = -\frac{Q\zeta(2)qr_+ \ln\left(\frac{r_+}{l}\right)}{\pi\sqrt{f'(r_+)}\hbar} + \frac{3\zeta(3)\sqrt{f'(r_+)}r_+}{8\pi^2}$$

$$f_1 = \frac{Q^3\zeta(2)a^2\sqrt{f'(r_+)}q^3r_+ \ln\left(\frac{r_+}{l}\right)}{8\pi\hbar^3} - \frac{3Q^2\zeta(3)a^2f'(r_+)^{\frac{3}{2}}q^2r_+}{64\pi^2\hbar^2} + \frac{Q\zeta(2)q \ln\left(\frac{r_+}{l}\right)}{\pi\sqrt{f'(r_+)}\hbar} +$$

$$+ \frac{Q\zeta(2)q}{\pi\sqrt{f'(r_+)}\hbar} - \frac{3\zeta(3)\sqrt{f'(r_+)}}{8\pi^2} \tag{A5}$$

$$f_2 = \frac{Q^3\zeta(2)a^2\sqrt{f'(r_+)}q^3 \ln\left(\frac{r_+}{l}\right)}{24\pi\hbar^3} + \frac{Q^3\zeta(2)a^2\sqrt{f'(r_+)}q^3}{24\pi\hbar^3} - \frac{Q^2\zeta(3)a^2f'(r_+)^{\frac{3}{2}}q^2}{64\pi^2\hbar^2} + \frac{Q\zeta(2)q}{3\pi\sqrt{f'(r_+)}\hbar r_+}$$

$$f_3 = \frac{Q^3\zeta(2)a^2\sqrt{f'(r_+)}q^3}{40\pi\hbar^3 r_+} .$$

## Appendix B. Coefficients $v_{kn}$ and $w_{kn}$

$$v_{00} = \left(-\frac{2Q^2\zeta(2)^2q^2 \ln(2\pi)^2}{\pi^3\left(\frac{A}{\pi l^2} - \frac{4\pi Q^2}{A}\right)} + \frac{2Q^2\zeta(2)^2q^2 \ln(2\pi)}{\pi^3\left(\frac{A}{\pi l^2} - \frac{4\pi Q^2}{A}\right)} + \frac{16Q^4\zeta(2)^4q^4 \ln(2\pi)^3}{3\pi^6A\left(\frac{A}{\pi l^2} - \frac{4\pi Q^2}{A}\right)^2}\right) \tag{A6}$$

$$v_{01} = \left(-\frac{3Q\zeta(2)\zeta(3)q \ln(2\pi)}{2\pi^4} + \frac{3Q\zeta(2)\zeta(3)q}{4\pi^4} + \frac{6Q^3\zeta(2)^3\zeta(3)q^3 \ln(2\pi)^2}{\pi^7A\left(\frac{A}{\pi l^2} - \frac{4\pi Q^2}{A}\right)}\right) \tag{A7}$$

$$v_{02} = \left(-\frac{9\zeta(3)^2\left(\frac{A}{\pi l^2} - \frac{4\pi Q^2}{A}\right)}{32\pi^5} + \frac{9Q^2\zeta(2)^2\zeta(3)^2q^2 \ln(2\pi)}{4\pi^8 A}\right), \quad v_{03} = \left(\frac{9Q\zeta(2)\zeta(3)^3q\left(\frac{A}{\pi l^2} - \frac{4\pi Q^2}{A}\right)}{32\pi^9 A}\right) \tag{A8}$$

$$v_{10} = \left(-\frac{2Q^2\zeta(2)^2q^2}{\pi^3\left(\frac{A}{\pi l^2} - \frac{4\pi Q^2}{A}\right)} + \frac{4Q^2\zeta(2)^2q^2 \ln(2\pi)}{\pi^3\left(\frac{A}{\pi l^2} - \frac{4\pi Q^2}{A}\right)} - \frac{16Q^4\zeta(2)^4q^4 \ln(2\pi)^2}{\pi^6A\left(\frac{A}{\pi l^2} - \frac{4\pi Q^2}{A}\right)^2}\right) \tag{A9}$$

$$v_{11} = \left( \frac{3Q\zeta(2)\zeta(3)q}{2\pi^4} - \frac{12Q^3\zeta(2)^3\zeta(3)q^3\ln(2\pi)}{\pi^7 A\left(\frac{A}{\pi l^2} - \frac{4\pi Q^2}{A}\right)} \right), \quad v_{12} = \left( -\frac{9Q^2\zeta(2)^2\zeta(3)^2q^2}{4\pi^8 A} \right), \quad v_{20} = \left( -\frac{2Q^2\zeta(2)^2q^2}{\pi^3\left(\frac{A}{\pi l^2} - \frac{4\pi Q^2}{A}\right)} + \frac{16Q^4\zeta(2)^4q^4\ln(2\pi)}{\pi^6 A\left(\frac{A}{\pi l^2} - \frac{4\pi Q^2}{A}\right)^2} \right) \quad \text{(A10)}$$

$$v_{21} = \left( \frac{6Q^3\zeta(2)^3\zeta(3)q^3}{\pi^7 A\left(\frac{A}{\pi l^2} - \frac{4\pi Q^2}{A}\right)} \right), \quad v_{30} = \left( -\frac{16Q^4\zeta(2)^4q^4}{3\pi^6 A\left(\frac{A}{\pi l^2} - \frac{4\pi Q^2}{A}\right)^2} \right), \quad v_{4k} = v_{5k} = 0. \quad \text{(A11)}$$

$$w_{00} = \left( -\frac{A\zeta(2)^2\ln(2\pi)^2}{8\pi^4} - \frac{Q^2\zeta(2)^4q^2\ln(2\pi)^4}{3\pi^7\left(\frac{A}{\pi l^2} - \frac{4\pi Q^2}{A}\right)} + \frac{Q^2\zeta(2)^4q^2\ln(2\pi)^3}{3\pi^7\left(\frac{A}{\pi l^2} - \frac{4\pi Q^2}{A}\right)} + \frac{8Q^4\zeta(2)^6q^4\ln(2\pi)^5}{5\pi^{10} A\left(\frac{A}{\pi l^2} - \frac{4\pi Q^2}{A}\right)^2} \right) \quad \text{(A12)}$$

$$w_{01} = \left( -\frac{3AQ\zeta(2)\zeta(3)q\left(\frac{A}{\pi l^2} - \frac{4\pi Q^2}{A}\right)\ln(2\pi)}{32\pi^5} - \frac{Q\zeta(2)^3\zeta(3)q\ln(2\pi)^3}{2\pi^8} + \frac{3Q\zeta(2)^3\zeta(3)q\ln(2\pi)^2}{8\pi^8} + \frac{3Q^3\zeta(2)^5\zeta(3)q^3\ln(2\pi)^4}{\pi^{11} A\left(\frac{A}{\pi l^2} - \frac{4\pi Q^2}{A}\right)} \right) \quad \text{(A13)}$$

$$w_{02} = \left( -\frac{9A\zeta(3)^2\left(\frac{A}{\pi l^2} - \frac{4\pi Q^2}{A}\right)^2}{512\pi^6} - \frac{9\zeta(2)^2\zeta(3)^2\left(\frac{A}{\pi l^2} - \frac{4\pi Q^2}{A}\right)\ln(2\pi)^2}{32\pi^9} + \frac{9\zeta(2)^2\zeta(3)^2\left(\frac{A}{\pi l^2} - \frac{4\pi Q^2}{A}\right)\ln(2\pi)}{64\pi^9} + \frac{9Q^2\zeta(2)^4\zeta(3)^2q^2\ln(2\pi)^3}{4\pi^{12} A} \right) \quad \text{(A14)}$$

$$w_{03} = \left( -\frac{9Q\zeta(2)\zeta(3)^3q\left(\frac{A}{\pi l^2} - \frac{4\pi Q^2}{A}\right)^2\ln(2\pi)}{128\pi^{10}} + \frac{9Q\zeta(2)\zeta(3)^3q\left(\frac{A}{\pi l^2} - \frac{4\pi Q^2}{A}\right)^2}{512\pi^{10}} + \frac{27Q\zeta(2)^3\zeta(3)^3q\left(\frac{A}{\pi l^2} - \frac{4\pi Q^2}{A}\right)\ln(2\pi)^2}{32\pi^{13} A} \right) \quad \text{(A15)}$$

$$w_{04} = \left( -\frac{27\zeta(3)^4\left(\frac{A}{\pi l^2} - \frac{4\pi Q^2}{A}\right)^3}{4096\pi^{11}} + \frac{81\zeta(2)^2\zeta(3)^4\left(\frac{A}{\pi l^2} - \frac{4\pi Q^2}{A}\right)^2\ln(2\pi)}{512\pi^{14} A} \right) \quad \text{(A16)}$$

$$w_{10} = \left( \frac{A\zeta(2)^2\ln(2\pi)}{4\pi^4} - \frac{Q^2\zeta(2)^4q^2\ln(2\pi)^2}{\pi^7\left(\frac{A}{\pi l^2} - \frac{4\pi Q^2}{A}\right)} + \frac{4Q^2\zeta(2)^4q^2\ln(2\pi)^3}{3\pi^7\left(\frac{A}{\pi l^2} - \frac{4\pi Q^2}{A}\right)} - \frac{8Q^4\zeta(2)^6q^4\ln(2\pi)^4}{\pi^{10} A\left(\frac{A}{\pi l^2} - \frac{4\pi Q^2}{A}\right)^2} \right) \quad \text{(A17)}$$

$$w_{11} = \left( \frac{3AQ\zeta(2)\zeta(3)q\left(\frac{A}{\pi l^2} - \frac{4\pi Q^2}{A}\right)}{32\pi^5} - \frac{3Q\zeta(2)^3\zeta(3)q\ln(2\pi)}{4\pi^8} + \frac{3Q\zeta(2)^3\zeta(3)q\ln(2\pi)^2}{2\pi^8} - \frac{12Q^3\zeta(2)^5\zeta(3)q^3\ln(2\pi)^3}{\pi^{11} A\left(\frac{A}{\pi l^2} - \frac{4\pi Q^2}{A}\right)} \right) \quad \text{(A18)}$$

$$w_{12} = \left( -\frac{9\zeta(2)^2\zeta(3)^2\left(\frac{A}{\pi l^2} - \frac{4\pi Q^2}{A}\right)}{64\pi^9} + \frac{9\zeta(2)^2\zeta(3)^2\left(\frac{A}{\pi l^2} - \frac{4\pi Q^2}{A}\right)\ln(2\pi)}{16\pi^9} - \frac{27Q^2\zeta(2)^4\zeta(3)^2q^2\ln(2\pi)^2}{4\pi^{12} A} \right) \quad \text{(A19)}$$

$$w_{13} = \left( \frac{9Q\zeta(2)\zeta(3)^3q\left(\frac{A}{\pi l^2} - \frac{4\pi Q^2}{A}\right)^2}{128\pi^{10}} - \frac{27Q\zeta(2)^3\zeta(3)^3q\left(\frac{A}{\pi l^2} - \frac{4\pi Q^2}{A}\right)\ln(2\pi)}{16\pi^{13} A} \right), \quad w_{14} = \left( -\frac{81\zeta(2)^2\zeta(3)^4\left(\frac{A}{\pi l^2} - \frac{4\pi Q^2}{A}\right)^2}{512\pi^{14} A} \right) \quad \text{(A20)}$$

$$w_{20} = \left( -\frac{A\zeta(2)^2}{8\pi^4} - \frac{2Q^2\zeta(2)^4q^2\ln(2\pi)^2}{\pi^7\left(\frac{A}{\pi l^2} - \frac{4\pi Q^2}{A}\right)} + \frac{Q^2\zeta(2)^4q^2\ln(2\pi)}{\pi^7\left(\frac{A}{\pi l^2} - \frac{4\pi Q^2}{A}\right)} + \frac{16Q^4\zeta(2)^6q^4\ln(2\pi)^3}{\pi^{10} A\left(\frac{A}{\pi l^2} - \frac{4\pi Q^2}{A}\right)^2} \right) \quad \text{(A21)}$$

$$w_{21} = \left( -\frac{3Q\zeta(2)^3\zeta(3)q\ln(2\pi)}{2\pi^8} + \frac{3Q\zeta(2)^3\zeta(3)q}{8\pi^8} + \frac{18Q^3\zeta(2)^5\zeta(3)q^3\ln(2\pi)^2}{\pi^{11} A\left(\frac{A}{\pi l^2} - \frac{4\pi Q^2}{A}\right)} \right) \quad \text{(A22)}$$

$$w_{22} = \left( -\frac{9\zeta(2)^2\zeta(3)^2\left(\frac{A}{\pi l^2} - \frac{4\pi Q^2}{A}\right)}{32\pi^9} + \frac{27Q^2\zeta(2)^4\zeta(3)^2q^2\ln(2\pi)}{4\pi^{12} A} \right), \quad w_{23} = \left( \frac{27Q\zeta(2)^3\zeta(3)^3q\left(\frac{A}{\pi l^2} - \frac{4\pi Q^2}{A}\right)}{32\pi^{13} A} \right) \quad \text{(A23)}$$

$$w_{30} = \left( -\frac{Q^2\zeta(2)^4 q^2}{3\pi^7\left(\frac{A}{\pi l^2} - \frac{4\pi Q^2}{A}\right)} + \frac{4Q^2\zeta(2)^4 q^2 \ln(2\pi)}{3\pi^7\left(\frac{A}{\pi l^2} - \frac{4\pi Q^2}{A}\right)} - \frac{16Q^4\zeta(2)^6 q^4 \ln(2\pi)^2}{\pi^{10} A\left(\frac{A}{\pi l^2} - \frac{4\pi Q^2}{A}\right)^2} \right) \quad \text{(A24)}$$

$$w_{31} = \left( \frac{Q\zeta(2)^3\zeta(3)q}{2\pi^8} - \frac{12Q^3\zeta(2)^5\zeta(3)q^3 \ln(2\pi)}{\pi^{11} A\left(\frac{A}{\pi l^2} - \frac{4\pi Q^2}{A}\right)} \right), \quad w_{32} = \left( -\frac{9Q^2\zeta(2)^4\zeta(3)^2 q^2}{4\pi^{12} A} \right) \quad \text{(A25)}$$

$$w_{40} = \left( -\frac{Q^2\zeta(2)^4 q^2}{3\pi^7\left(\frac{A}{\pi l^2} - \frac{4\pi Q^2}{A}\right)} + \frac{8Q^4\zeta(2)^6 q^4 \ln(2\pi)}{\pi^{10} A\left(\frac{A}{\pi l^2} - \frac{4\pi Q^2}{A}\right)^2} \right), \quad w_{41} = \left( \frac{3Q^3\zeta(2)^5\zeta(3)q^3}{\pi^{11} A\left(\frac{A}{\pi l^2} - \frac{4\pi Q^2}{A}\right)} \right), \quad w_{50} = \left( -\frac{8Q^4\zeta(2)^6 q^4}{5\pi^{10} A\left(\frac{A}{\pi l^2} - \frac{4\pi Q^2}{A}\right)^2} \right) \quad \text{(A26)}$$

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
