# Peer review of "Noncommutative Correction to the Entropy of Charged BTZ Black Hole"

_symmetry, doi:10.3390/sym15020417_

Round 1
Reviewer 1 Report
See the attachment

Reviewer 2 Report
cxThis is a good and interesting paper, concerning the noncommutative version of the charged BTZ black hole. The authors claim to compute the entropy of this black hole. This paper is generally very well written and well organized, but I have some serious doubts, whether it is mathematically correct?
The authors make some approximations and then the corrections should have the same mass dimension, or length dimension, as the object itself. See for instance equation (105): There are three equations and I claim that only the middle equation is mathematically correct and two others are incorrect.
Take for instance the third equation in equation (105): Here the object is clearly dimensionless, but the correction clearly has the length dimension length squared. The correction should of course be $r/l$ squared, or something like that, so that it is dimensionless. The same for the first equation in equation (105). Etc etc.
Further revision is necessary. Major revision.
Reviewer 3 Report
Please see the attached pdf file: Report--symmetry-2146066

Round 2
Reviewer 2 Report
The authors of this paper, have answered my question and comment, which I raised in my first report, in a very satisfying way and they have improved their paper in many different ways, so I can now clearly recommend this paper for publication in this journal.
The authors of this paper, have answered my question and comment, which I raised in my first report, in a very satisfying way and they have improved their paper in many different ways, so I can now clearly recommend this paper for publication in this journal.